# The Divergent Key Residues of Two *Agrobacterium fabrum* (*tumefaciens*) CheY Paralogs Play a Key Role in Distinguishing Their Functions

**DOI:** 10.3390/microorganisms9061134

**Published:** 2021-05-24

**Authors:** Dawei Gao, Renjie Zong, Zhiwei Huang, Jingyang Ye, Hao Wang, Nan Xu, Minliang Guo

**Affiliations:** College of Bioscience and Biotechnology, Yangzhou University, Yangzhou 225009, China; DX120180135@yzu.edu.cn (D.G.); rjiezong@163.com (R.Z.); huangzhiwei@hyit.edu.cn (Z.H.); snowsheep2011@163.com (J.Y.); wanghao@yzu.edu.cn (H.W.); nanxu@yzu.edu.cn (N.X.)

**Keywords:** *Agrobacterium fabrum*, chemotaxis, cost attractant, paralog function, residue substitution, response regulator

## Abstract

The chemotactic response regulator CheY, when phosphorylated by the phosphoryl group from phosphorylated CheA, can bind to the motor switch complex to control the flagellar motor rotation. *Agrobacterium fabrum* (previous name: *Agrobacterium tumefaciens*), a phytopathogen, carries two paralogous *cheY* genes, *cheY1* and *cheY2*. The functional difference of two paralogous CheYs remains unclear. Three *cheY*-deletion mutants were constructed to test the effects of two CheYs on the chemotaxis of *A.*
*fabrum*. Phenotypes of three *cheY*-deletion mutants show that deletion of each *cheY* significantly affects the chemotactic response, but *cheY2*-deletion possesses more prominent effects on the chemotactic migration and swimming pattern of *A. fabrum* than does *cheY1*-deletion. CheA-dependent cellular localization of two CheY paralogs and in vitro pull-down of two CheY paralogs by FliM demonstrate that the distinct roles of two CheY paralogs arise mainly from the differentiation of their binding affinities for the motor switch component FliM, agreeing with the divergence of the key residues on the motor-binding surface involved in the interaction with FliM. The single respective replacements of key residues R93 and A109 on the motor-binding surface of CheY2 by alanine (A) and valine (V), the corresponding residues of CheY1, significantly enhanced the function of CheY2 in regulating the chemotactic response of *A. fabrum* CheY-deficient mutant Δy to nutrient substances and host attractants. These results conclude that the divergence of the key residues in the functional subdomain is the decisive factor of functional differentiation of these two CheY homologs and protein function may be improved by the substitution of the divergent key residues in the functional domain for the corresponding residues of its paralogs. This finding will help us to better understand how paralogous proteins sub-functionalize. In addition, the acquirement of two CheY2 variants, whose chemotactic response functions are significantly improved, will be very useful for us to further explore the mechanism of CheY to bind and regulate the flagellar motor and the role of chemotaxis in the pathogenicity of *A. fabrum*.

## 1. Introduction

Chemotaxis allows the motile microbes to move towards more favorable environments via sensing and tracking the gradients of attractants and repellents in their surroundings. Chemotaxis signal is transduced by a two-component system, which typically consists of a central histidine kinase CheA with a sensor domain and a phosphorylatable response regulator CheY [1,2,3,4]. Chemotaxis signal transduction begins with the recognition of chemoeffectors by chemoreceptors (also called methyl-accepting chemotaxis protein, MCP) that form a ternary core complex with the coupling protein, CheW, and the sensor kinase, CheA [5,6]. In the model bacterium, *E. coli*, the MCP-CheW-CheA ternary core complexes are arrayed in cell poles. Generally, when chemoeffectors bind to the ligand binding domain of the transmembrane MCPs, the cytoplasmic signaling domain of MCP transduces the signal from chemoeffectors to CheA under the mediation of the coupling protein CheW and regulates the activity of CheA. The histidine kinase CheA can auto-phosphorylate and then transfer the phosphoryl group to CheY. The phosphorylated CheY (CheY-P) regulates the flagellar motor rotation via binding to two components (FliM and FliN) of the motor switch complex at the bottom of flagellum [7,8]. The phosphoryl group of CheY-P can be removed by CheZ, a phosphorylated CheY-specific phosphatase, and then the signal is rapidly terminated [9,10].

Unlike *E. coli*, which has only one set of chemosensory systems and encodes only one *cheY* for chemotaxis, many bacteria usually have more than one chemosensory system and multiple response regulator CheYs [11]. Not all CheY paralogs participate the chemotactic signal transduction. For example, *Borrelia burgdorferi* possesses two complete sets of core chemotaxis proteins and three CheYs. Only CheY3 is essential for motility and chemotaxis [12]. CheY2 is suggested to regulate a virulence determinant that is required for the infectious life cycle of *Borrelia burgdorferi* [13]. *Vibrio cholerae* has three sets of chemotaxis proteins and encodes five CheYs, but only one of the five CheY paralogs directly switches flagellar rotation. The remaining CheYs are suggested to control functions other than chemotaxis [14]. *Azospirillum brasilense* encodes seven CheY paralogs. CheY6 and CheY7 play a role in controlling swimming reversals, CheY4 regulates flagellar motor pauses, and CheY1 controls the transient change of swimming speed [15,16]. Some bacteria have only one complete set of core chemotaxis proteins, but two or more CheYs. For example, both *Rhizobium* (now, *Sinorhizobium* or *Ensifer*) *meliloti* [17] and *Azorhizobium caulinodans* [18] have only one complete set of core chemotaxis proteins, but two CheYs. The phenotypes of *cheY*-deletion mutants of both bacteria show that both CheYs are required for full tactic response with one CheY having a more prominent role than the other CheY.

*Agrobacterium fabrum* is a soil-born phytopathogen that causes tumor disease on various dicotyledonous plants [19,20]. It is a motile α-proteobacterium with a highly sensitive chemotaxis system. The genome of *A. fabrum* C58 carries only one complete set of core chemotaxis genes. Compared with most bacteria, *A. fabrum* chemotaxis system possesses three significant features: (a) most core chemotaxis genes (except for CheW-encoding genes and MCP-encoding genes) are in one gene cluster; (b) *A. fabrum* genome encodes two coupling proteins (CheW1 and CheW2) and two response regulators (CheY1 and CheY2); (c) it lacks the CheZ-encoding gene, but encodes an additional CheS protein; (d) and *A. fabrum* genome encodes 20 MCPs, much more than many bacteria [21]. The gene organization of the chemotaxis cluster in *A.*
*fabrum* is very similar to that in the closely related species, *Sinorhizobium meliloti* [9,22] and *Rhizobium leguminosarum* [22,23]. It was reported that two *S. meliloti* CheYs play clearly different roles in transducing the chemotactic signal. An early investigation on the phosphotransfer between the purified recombinant CheY1, CheA and CheY2 showed that there is a rapid phosphotransfer from CheY2-P via CheA to CheY1 in the three-component mixture, CheY1/CheA/CheY2, suggesting that CheY2 is the genuine chemotaxis response regulator and CheY1 acts as a phosphate sink [24]. CheY1 is indirectly involved in the chemotactic signal transduction through regulating the termination of chemotactic signal [17]. Later it was found that chemotaxis protein CheS significantly increases the dephosphorylation of CheY1-P but not CheY2-P and the affinity of CheY1 to CheA/CheS is ~100-fold stronger than that to CheA. Therefore, it was proposed that CheS is required for facilitating the chemotactic signal termination via promoting CheY1-P dephosphorylation [25]. However, it is unclear whether two *A. fabrum* CheY paralogs play different roles in responding and transducing the chemotaxis signal. In this article, we aim at exploring the function of two *A. fabrum* CheY paralogs.

## 2. Materials and Methods

### 2.1. Bacterial Strains, Plasmids, Primers, and Bacterial Growth Conditions

*E. coli* strains, derivatives of *A. fabrum* C58 and plasmids used in this work are described in Appendix A. The used primers and their sequences are listed in Appendix A. *E. coli* strains were grown in lysogeny broth liquid medium at 37 °C with shaking (210 rpm) [26]. *A. fabrum* strains were grown in MG/L liquid medium at 28 °C with shaking (200 rpm) [27,28,29]. The composition of the corresponding solid medium is the same as the liquid, except for the extra 1.5% of agar. When necessary, antibiotics were added to the final concentrations: for *E. coli*, ampicillin (Ap) with 100 µg/mL, kanamycin (Km) with 50 µg/mL; for *A. fabrum*, kanamycin with 100 µg/mL, and carbenicillin (Cr) with 100 µg/mL [30,31].

### 2.2. DNA Manipulation, Construction of Gene-Deleted Mutants and Complemented Strains

The standard molecular protocols were used for DNA manipulation [26]. *A. fabrum* gene sequences refer to the published genome sequence (genome accession number: AE007869.2) [32,33]. The TIANprep Mini Plasmids Kit (TIANGEN BIOTECH Corporation, Beijing, China) was used for the purification of plasmid DNA. The TaKaRa MiniBEST Agrose Gel DNA Extraction Kit Ver 4.0 (TaKaRa Corporation, Dalian, China) was used for the purification of PCR products and DNA fragments. *E. coli* competent cells were purchased from the Weidibio Company (Shanghai, China). Restriction endonucleases were purchased from the Takara Corporation. Ligation-Free Cloning System (ABM Corporation, Vancouver, Canada) was used for the construction of recombinant plasmids.

The precise in-frame deletions of *A. fabrum* genes, *cheY1*, *cheY2* and *cheS* were constructed according to the gene replacement procedure as described previously [34,35]. pEX18Km, which carries a positive selection marker (kanamycin resistant gene) and a counter-selectable marker (suicide gene *sacB*), was used for constructing gene replacement plasmids [36]. Plasmids were transferred to *A. fabrum* cells by electroporation [37]. The desired *A. fabrum* mutants were screened by PCR and verified by DNA sequencing as described previously [35].

The complemented strains of *cheY*-deficient mutants were constructed by introducing a *cheY*-expressing plasmid to the *cheY*-deficient mutants. To construct two *cheY*-expressing plasmids, *cheY1* and *cheY2* genes were amplified through PCR from *A. fabrum* genomic DNA and were cloned into plasmid pUCA-19 under the control of *lacZ* promoter, respectively. To construct the plasmid that can express both CheY1 and CheY2, both *cheY1* and *cheY2* genes were cloned into plasmid pUCA-19 and were controlled by the same *lacZ* promoter. Thus, three complementing plasmids (pUCA-Y1, pUCA-Y2, and pUCA-Y12) were constructed. Plasmids pUCA-Y1 and pUCA-Y2 were used to complement *cheY1* and *cheY2*, respectively, and plasmid pUCA-Y12 was used to complement both of *cheY1* and *cheY2* simultaneously in the same cell.

### 2.3. Chemotaxis Assay

Chemotactic response of *A. fabrum* to nutrient substances was tested in the AB-sucrose swim plate containing 0.2% Bacto agar (Solarbio, Beijing, China) according to previous studies [27,38]. Briefly, cells of various *A. fabrum* strains were collected from the cultures in the middle-logarithmic growth phase, washed twice with fresh AB-sucrose liquid medium, and resuspended in AB-sucrose liquid medium. Cell concentration of every cell suspension was normalized to 5 × 10^8^ cfu/mL. An equal volume (3 μL) of various cell suspensions was inoculated on the swim agar plate and the distances between inoculating spots were equal. For the test of the chemotactic response of *A. fabrum* to host plant attractants, a fresh leaf disc (diameter = 5 mm) of kalanchoe was placed at the center of the swim agar plate and various strains were inoculated onto the plate surface 2.5 cm from the leaf disc [27]. The plates inoculated with these strains were incubated at 28 °C for 40 h without disturbance and observed occasionally during the incubation period. Diameter of the colony was used to quantify the chemotaxis to nutrient substances. Relative migration distance of the colony, which was defined as the distance from the inoculating spot to the edge toward the leaf disc minus the distance from the inoculating spot to the edge away from the leaf disc, was used to quantify the chemotaxis to plant attractants. Chemotactic response to single attractant (acetosyringone (AS) or sucrose) was determined by a capillary assay as described in previous works [27]. The final concentrations of AS and sucrose in the capillary tube were 10^−7^ and 10^−6^ mol/L, respectively.

### 2.4. Observation and Analysis of Swimming Behavior

To track the movement of bacteria, an eGFP-expressing plasmid was introduced into each strain. Cells in the middle-logarithmic growth phase were diluted with buffer (EDTA at 0.1 mmol/L and KH_2_PO_4_ at 10 mmol/L, pH 7.0) until clear and countable cells were observed under a fluorescence microscope. The movement of free-swimming cells were recorded using the 100 × objective lens of an inverted fluorescence microscope (Mshot, Guangzhou, China) equipped with a digital camera (model MS23) at a rate of 40 frames per second. For each strain, at least 50 individual cells were recorded for up to 15 s. The swimming behavior of the cells recorded by the videos were analyzed by using ICY software. Two swimming parameters, swimming velocity and reorientation frequency, were used to characterize the swimming behavior of *A. fabrum*. Velocity was expressed as the smooth-swimming distance per second (μm/s), and reorientation frequency was defined as the number of swimming reorientations per second and per cell [4,14,15]. Values of velocity and reorientation frequency were calculated from a minimum of 50 cells recorded from at least three different fields of view.

### 2.5. Construction of Plasmids for Expressing the Split-eGFP Fusion Proteins

To test the interaction between CheA and CheY in vivo, eGFP protein was split at the peptide bond between amino acid residues 158 and 159 [39]. The C-terminal part of the eGFP protein (residues 158–238) was fused to the C-terminus of CheA with a seven-residue linker (GTSGGSG) between two fused parts to generate CheA-linker-C*egfp* fusion protein. The N-terminal part of the eGFP protein (residues 2–157) was fused to the C-terminus of CheY1 with an eight-residue linker (GGSGSGSR) to generate CheY1-linker-N*egfp* fusion protein. To ensure that both fusion proteins have the same expression level, the fusion genes encoding two fusion proteins were cloned into the same plasmid and controlled by the same promoter (*lacZ* promoter). The expression cassette for these two fusion proteins was organized as *promoter-SD-cheA-lin-Cegfp-stop-SD-cheY1-lin-Negfp-stop* (*SD*: SD sequence for ribosome binding; *lin*: linker-encoding sequence; *stop*: stop code), and inserted into plasmid pUCA19 to generate plasmid pUCA-SGAY1. This plasmid was introduced into strain Δay for the equivalent expression of two fusion proteins (CheA-C*egfp* and CheY1-N*egfp*) in the triple-deletion mutant. The design similar to that of plasmid pUCA-SGAY1 was used to construct the following plasmids for the equivalent expression of other (fusion) proteins: pUCA-SGAY2 for the equivalent expression of CheA-C*egfp* and CheY2-N*egfp*; pUCA-SGAY1-Y2 for the equivalent expression of CheA-C*egfp*, CheY1-N*egfp*, and native CheY2; pUCA-SGAY2-Y1 for the equivalent expression of CheA-C*egfp*, CheY2-N*egfp*, and native CheY1; pUCA-SGY12 for the equivalent expression of CheY2-C*egfp* and CheY1-N*egfp*; pUCA-SGAW1 for the equivalent expression of CheA-C*egfp* and CheW1-N*egfp* (as a positive control, Appendix A); pUCA-SGY1 for the equivalent expression of C*egfp* and CheY1-N*egfp* (as a negative control).

### 2.6. Microscopic Observation of the Reconstituted Split-eGFP Fusions

Agrobacterial strains that can express the split-eGFP fusion proteins were grown to the middle-logarithmic growth phase. A droplet of cell culture was added onto the center of the slide and covered by a coverslip. To prevent the cell culture from drying, acrylic polymer was used to seal four edges of the coverslip. Images were taken with a confocal laser scanning microscope (Leica TCS SP8 STED 3X system) with a 100× oil-immersion objective. Fluorescence of eGFP was excited by Ar laser (excitation wavelength 488 nm) and detected by using a bandpass 505–550 nm (emission wavelength) filter.

### 2.7. Protein Overexpression and GST Pull-Down Assay

Genes encoding CheY1 (*atu0516*), CheY2 (*atu0520*) and FliM (*atu0561*) were amplified from the genomic DNA of *A. fabrum* strain C58. *cheY1* and *cheY2* were cloned into pET30a vector with a C-terminal 6-histidine tag, respectively. *fliM* were cloned into the pGEX-4T-1 expression vector with an N-terminal GST tag, respectively. Recombinant proteins were expressed in *E. coli* strain BL21 (DE3). *E. coli* cells harboring various fusion protein-expressing plasmids were cultured to an OD_600 nm_ of 0.4 to 0.6 at 37 °C and then 0.6 mM IPTG (isopropyl-β-D-thiogalactopyranoside) was added to the culture to induce the expression of fusion protein. The cells were grown for an additional 4 h at 25 °C with shaking (150 rpm). Cells from 20 mL of culture were collected by centrifugation at 5000× *g* and 4 °C for 5 min. After washed twice, the cells were re-suspended in 1 mL of phosphate-buffered saline (PBS: 1.8 mM KH_2_PO_4_, 10 mM Na_2_HPO_4_, 2.7 mM KCl, 140 mM NaCl) containing 60 μL of 1M MgCl_2_, 10 μL of APMSF (4-amidino phenyl methane sulfonyl fluoride), and 100 μL of either 0.5 M acetyl phosphate (phosphorylating conditions) or water (non-phosphorylating conditions) [8]. Cells in the suspension were disrupted by sonication on ice. Cell debris was removed by centrifugation (12,000× *g*, 15 min and 4 °C). The supernatants from the *E. coli* cells expressing various fusion proteins were the crude extracts (or preparations) of the corresponding fusion proteins. The concentrations of CheY1-6×his and CheY2-6×his in their respective crude extracts were estimated by Western blotting using the anti-6×his-tag antibody and then adjusted to the same with PBS for the GST pull-down assay (Appendix A). An amount of 1 mL of the crude extract of GST or GST-FliM was mixed with 160 μL of a 50% slurry of glutathione-Sepharose 4B beads (BEYOTIME BIOTECH, China) prepared according to the user’s manual. After incubation at 4 °C for 1 h with gentle rocking, the beads binding with GST or GST-FliM were pelleted from the slurry by centrifugation (1000g, 10 s and 4 °C) and washed five times with PBS. Equal amounts of the GST fusion protein-bound beads were mixed with equal amounts of the crude extract of CheY1-6×his or CheY2-6×his and incubated at 4°C for 1 h with rocking to allow the binding of CheY1 or CheY2 to the FliM on the beads. The beads binding with the complex of CheY1 (or CheY2)- FliM were washed twice with PBS, boiled with equal amount of SDS-PAGE loading buffer, and used for the SDS-PAGE and immunoblotting analyses.

### 2.8. Electrophoresis and Immunoblotting Analysis

Electrophoresis and immunoblotting analyses were carried out as described before [27,40]. Equal volumes (10 μL) of various samples were loaded for SDS-PAGE. For immunoblotting, proteins were transferred to polyvinylidene difluoride membranes (Merck Millipore, Darmstadt, Germany) and were detected with the BeyoECL Star kit (Beyotime biotechnology Corp., China) using the automatic chemiluminescence image analysis system Amersham Imager 600 (General Electric Corp., Boston, MA, USA). Monoclonal antibody against the His-tag sequence region was used as the primary antibody for the detection of the fusion proteins of CheY1-6×his and CheY2-6×his.

### 2.9. Site-Directed Mutation of CheY-Encoding Gene and the Test of the Function of Different CheY Variants 

DNAMAN 8.0 (Lynnon Biosoft, San Ramon, CA, USA) was used for protein sequence alignment. The amino acid sequences of two agrobacterial CheY proteins were aligned with the CheY of *E. coli* (Appendix A). The amino acid residues for the substitution were chosen according to the sequence alignment and the previous research in *E. coli* CheY [41,42,43,44]. Site-directed mutation was operated by using the TaKaRa MutanBEST Kit according to the manual provided by the manufacturer. Genes encoding various CheY variants were respectively cloned into pUCA-19 to generate the corresponding plasmids (Appendix A). These CheY variant-expressing plasmids were respectively introduced into the CheY-deficient *A. fabrum* strain Δy by electroporation to test whether these CheY variants could complement the function of CheY in the chemotactic response of *A. fabrum*.

### 2.10. Statistical Analysis 

The statistical analysis in this article was performed using Microsoft Office Excel’s data analysis tool (2019 version). Non-normally-distributed data (the data of reorientation frequency in this research) were conducted with the Wilcoxon–Mann–Whitney test. Normally-distributed data (the other data in this research) were conducted using the unpaired Student’s *t*-test. The *p*-value was used to assess the statistical difference between the measurements.

## 3. Results

### 3.1. Deletion of Either cheY Gene Significantly Affects the Chemotactic Migration of A. fabrum in Swim Plate with cheY2 having Prominent Effect

The genome of *A. fabrum* carries two paralogous *cheY* genes, *cheY1* (*atu0516*) and *cheY2* (*atu0520*). Both are in the unique chemotaxis operon of *A. fabrum* [9]. To identify the effects of two *cheY* genes on the chemotaxis of *A. fabrum*, three *cheY*-deletion mutants (Δy1: *cheY1*-deletion mutant; Δy2: *cheY2*-deletion mutant; Δy: *cheY1*-*cheY2* double-deletion mutant) were constructed by using a precise gene replacement system. Meanwhile, four complemented strains (Δy1 + y1: Δy1 complemented by *cheY1*; Δy2 + y2: Δy2 complemented by *cheY2*; Δy + y1: Δy complemented by *cheY1*; Δy + y2: Δy complemented by *cheY2*) were constructed by the introduction of *cheY*-expressing plasmid. The mutant Δa, which lacks CheA, the central player of the chemotaxis system, was used as the negative control of chemotaxis. All these mutant strains were inoculated on the same swim plate to compare their chemotactic migrations. When bacterial cells are grown on the plate, they consume the nutrient substances around the inoculating spot, resulting in the concentration gradient of nutrient substances. If bacterial cells have a sensitive chemotactic response, they move outward along the nutrient gradient and grow large colonies. Otherwise, their colonies will be small. Therefore, the colony size can be used to characterize their chemotactic migrations [27,38]. The colony sizes of different *A. fabrum* strains are shown in Figure 1. The data demonstrate that: (a) both CheYs are required for the full chemotactic response of *A. fabrum* to nutrient gradient; (b) CheY1 and CheY2 have significantly different effects on the chemotactic response of *A. fabrum*, with CheY2 being more important than CheY1 for the chemotactic response. These results are consistent with a previous study on *A. fabrum* [45] and the different effects of two CheYs on the chemotactic response of *A. fabrum* are similar to those of *Rhizobium meliloti* [17] and *Azorhizobium caulinodans* [18]. The complementation of CheY1 and CheY2 to the corresponding CheY-deficient mutants (Δy1 and Δy2) confirms that the mutants were constructed correctly and did not affect the normal expression of other genes.

### 3.2. CheY1 and CheY2 have Different Effects on the Swimming Behavior of A. fabrum

Since CheY1 and CheY2 have different effects on the chemotactic migration of *A. fabrum* in the swim plate, we want to know if two CheYs have different effects on the swimming behavior of *A. fabrum*. Usually, swimming velocity and reorientation frequency are used to characterize the swimming behavior [4,15]. Therefore, free-swimming cells of every strain were tracked by video. Their swimming velocity was calculated in micrometers per second (μm/s) and the frequency of reorientation was measured as the number of changes in swimming direction per second and per cell. For better tracking the swimming patterns of every *A. fabrum* strain, a plasmid carrying a fluorescent reporter gene (*gfp*) was introduced into every mutant to express GFP, to obtain a timely recording of the motility of agrobacterial cells by the fluorescence microscope system.

For many bacteria [14,15,17,18], losing the control of CheY usually makes bacterial cells swim faster with fewer reorientations. As shown in Figure 2, the deficiency of CheY2 (Δy2) causes a significant increase in the smooth-swimming velocity, but significantly suppresses the reorientation of the swimming direction. The effects of CheY1-deficiency on the smooth-swimming velocity and the frequency of reorientation are statistically insignificant, although the CheY1-deficient mutant (Δy1) shows a slight increase in the smooth-swimming velocity and a slight reduction in the ability of reorientation compared with the wild-type strain (C58). Results of the CheY-complemented strains further verify that the complementation of CheY2 affects the smooth-swimming velocity and reorientation frequency of CheY-deficient mutant significantly, but the effect of CheY1-complementation on the smooth-swimming velocity and reorientation frequency of CheY-deficient mutant is insignificant. Therefore, the difference between the effects of two CheYs on the swimming behavior of *A. fabrum* is significant and CheY2 is the key chemotactic response regulator regulating the swimming behavior of *A.*
*fabrum*.

### 3.3. Both CheY1 and CheY2 Interact with CheA at the Cell Pole Respectively

As we know, MCPs are located at the cell poles and CheW couples CheA to MCPs to form the stable MCP-CheW-CheA ternary complex [10]. CheY, as the response regulator of chemotaxis, can bind to CheA and be phosphorylated by the phosphoryl group from the phosphorylated CheA [8]. If both CheY1 and CheY2 of *A. fabrum* were involved in the chemotaxis signaling transduction, the co-localization of CheA and CheY1 (or CheY2) at cell poles could be observed. To visualize the co-localization of CheA and CheY1 (or CheY2) in the *A. fabrum* cell, eGFP was split into two non-fluorescent parts (N-terminal part and C-terminal part) at the peptide bond between residues 158 and 159 [39]. The C-terminal part (C*egfp*) of eGFP was fused to the C-terminus of CheA to form the CheA-C*egfp* fusion protein and the N-terminal part (N*egfp*) of eGFP was fused to the C-terminus of CheY1 (or CheY2) to form the CheY1-N*egfp* (or CheY2-N*egfp*) fusion protein. The interaction between CheA and CheY1 (or CheY2) brings two split-eGFP parts together and makes the reconstituted eGFP emit fluorescence. To eliminate the possible effects of the in-situ CheA, CheY1 and CheY2 on the interactions between these fusion proteins, the fusion protein-expressing plasmid was introduced into the *cheA*-*cheY1*-*cheY2* triple-deletion mutant (Δay). The *cheY1*-*cheY2* double-deletion mutant (Δy) carries a plasmid, which can express the CheY1-Negfp fusion protein and C*egfp* (C-terminal part of eGFP), was used as the negative control. As shown in Figure 3, clear polar fluorescence focus was observed in *A. fabrum* cells expressing two pairs of fusion proteins (CheA-C*egfp* paired with CheY1-N*egfp* and CheA-C*egfp* paired with CheY2-N*egfp*), respectively. Compared with CheA-C*egfp*/CheY2-N*egfp*, CheA-C*egfp*/CheY1-N*egfp* produced a higher percentage of polar fluorescence and the presence of one native CheY could significantly prohibit the other CheY fusion protein from interacting with CheA-C*egfp*. These results demonstrate that CheY1 and CheY2 interact with CheA competitively and the affinity of CheY1 to CheA is higher than that of CheY2. The absence of CheS slightly affected the interactions of both CheY1 and CheY2 with CheA. Figure 3 also showed that there was no interaction between two CheY proteins.

### 3.4. Two CheY Proteins Show Significantly Different Affinities to FliM

CheY1 and CheY2 show significantly different effects on the chemotactic migration and swimming behavior of *A. fabrum*. Next, we aimed to explore the difference of two CheYs in the interaction with FliM, one component of the switch complex on the basal body of flagellar motor. According to our current knowledge of CheY, CheY-P can bind to FliM with significantly higher affinity than CheY, but the half-life of CheY-P is very short, and easy to autodephosphorylate [4,24,46]. Therefore, acetyl phosphate was used as the CheY-phosphorylating agent to phosphorylate CheY and to keep CheY in the phosphorylation state [8]. The interaction between CheY and FliM was tested by pull-down assay. FliM of *A. fabrum* was expressed as GST-FliM fusion protein. CheY1 and CheY2 of *A. fabrum* were expressed as CheY1-6×his and CheY2-6×his fusion proteins, respectively. GST-FliM on the glutathione resin served as bait to bind CheY1-6×his (or CheY2-6×his) in the presence, or absence of acetyl phosphate. GST were used as the negative control of the pull-down assay. The pulled-down proteins were separated by SDS-PAGE and analyzed by Western blotting with antibody against the 6×his tag present in both CheY1 and CheY2. The band intensity was quantified densitometrically and used to estimate the amount of CheY1-6×his (or CheY2-6×his) pulled down by GST-FliM (or GST) (Appendix A). As shown in Figure 4, the band intensity of CheY2-6×his pulled down by GST-FliM is ~5 fold higher than the band of CheY1-6×his pulled down by GST-FliM, demonstrating that the affinity of FliM for CheY2 is significantly higher than that for CheY1. Results in Figure 4 also show that GST-FliM barely pulled down either of two CheYs in the absence of acetyl phosphate.

### 3.5. The Single Respective Replacements of R93 and A109 of CheY2 by the Residues of the Corresponding Sites in CheY1, Significantly Enhance the Function of CheY2 in the Absence of CheY1

Since the affinities of CheY1 and CheY2 for FliM are different, we emphasized the difference between CheY1 and CheY2 in the key residues involved in the interaction with FliM. Key residues in the motor binding surface of *E. coli* CheY, which are involved in the interaction with FliM, include Ala90, Lys92, ILe95, Ala99, Tyr106, Val108, Lys119 and Lys122 [42]. We aligned the sequences of two *A. fabrum* CheYs with the sequence of *E. coli* CheY. As shown in Appendix A, the corresponding eight key residues of CheY1 are S86, A88, K91, R95, W102, V104, A115 and R118 respectively. Residues of CheY2 in these eight corresponding sites are G91, R93, V96, A100, V107, A109, A120, and A123 respectively. Seven key residues in the motor binding surface of two *A. fabrum* CheYs are divergent. We chose six key residues (corresponding to K92, I95, A99, Y106, V108, and K122 of *E. coli* CheY) to conduct the residue substitution. To test how these key residues affect the functions of two CheY proteins in the chemotactic response of *A. fabrum*, these six residues were exchanged between the two CheYs in the form of single respective substitutions. All these single residue-substituted CheY variants were used to complement the CheY-deficient mutant Δy, respectively. The activities of these CheY variants in restoring the chemotactic response of *A. fabrum* CheY-deficient mutant were tested by using the swim plate assay. Results show that all the single residue-substituted CheY1 variants are similar to native CheY1 in restoring the chemotactic migration of Δy (Figure 5A,B), whereas, different single residue-substituted CheY2 variants show different activities in restoring the chemotactic migration of Δy. The activities of two CheY2 variants, CheY2^R93A^ and CheY2^A109V^, are significantly higher than that of native CheY2 in restoring the chemotactic migration of Δy (Figure 5C,D).

In addition, we further tested the effects of these two CheY2 variants on the chemotactic response of *A. fabrum* to host plant attractants. The chemotactic response of *A. fabrum* to host attractants was tested by the attraction of agrobacterial cells to the disc of kalanchoe leaf [27,47]. The kalanchoe leaf disc on the center of the swim plate can steadily produce exudates, which include host attractants, and creates a steady host attractant gradient. Agrobacterial cells that have chemotactic function will be attracted by the attractants and move toward the leaf disc along the host attractant gradient. The chemotactic response to host attractants was characterized by the migration distance difference of the colony between the edges toward and away from the leaf disc [27]. As shown in Figure 6A,B, these two CheY2 variants significantly enhance the chemotactic response of *A. fabrum* Δy mutant to host attractants, in comparison with the native CheY2. The chemotactic responses of *A. fabrum* to two single attractants, sucrose and acetosyringone (AS), were tested by a capillary assay as before [27]. The effects of various CheY2 variants on the chemotactic responses of *A. fabrum* to two single attractants are similar to those to host exudates (Figure 6C,D), confirming that two single residue substitutions (R93A and A109V) significantly improve the function of CheY2 in the absence of CheY1.

## 4. Discussion

*A. fabrum* genome encodes one complete set of core chemotaxis components, two CheWs and two CheYs. Our previous investigation showed that both CheW1 and CheW2 are required for full chemotactic behavior of *A.*
*fabrum* but neither of them is indispensable to chemotaxis [27]. In this study, we examined the roles of two *A. fabrum* CheY paralogs in regulating the chemotactic response and swimming behavior of *A. fabrum*. Phenotypes of *A. fabrum cheY*-deletion mutants show that two CheY paralogs have significantly different effects on the chemotactic response of *A. fabrum* and CheY2 plays a key role in mediating the chemotactic response. Observation of the free-swimming behaviors of *A. fabrum cheY*-deletion mutants also shows that CheY2 is the major regulator of the free-swimming behavior. The phenotypes of the chemotactic response and free-swimming behavior of *A. fabrum cheY*-deletion mutants are very similar to those of *S. meliloti cheY*-deletion mutants [17].

Our observation of the co-localization of CheA and CheY finds that both CheYs are competitively co-localized with CheA at the cell pole and CheS has slight effect on the co-localization of CheA with two CheYs. Both the effect of one CheY on the other CheY co-localized with CheA and the effect of CheS on two CheYs co-localized with CheA are within the same order of magnitude range. This is quite different from *S. meliloti*, where only CheY2 was observed to be co-localized with CheA at a cell pole, but CheY1 was not [48]. Meanwhile, the absence of CheS decreased the binding of *S. meliloti* CheY1 to CheA almost two orders of magnitude, but the effect of CheS on the binding of *S. meliloti* CheY2 to CheA was within the same order of magnitude range [25]. Our results show that the affinity of CheY2 to FliM is approximately 5-fold higher than that of CheY1 (Figure 4). The difference between two CheY’s affinities to FliM is significantly higher than that between two CheY’s affinities to CheA, implying that the functional difference between two *A. fabrum* CheY paralogs lies mainly in their interactions with FliM, not with CheA. Combining these results, we can conclude that the mechanism resulting in the functional difference of two *A. fabrum* CheY paralogs is different from that of two *S. meliloti* CheY paralogs, although the phenotypes of *cheY*-deficient mutants of both bacteria are similar and the sequences of *S. meliloti* CheY1 and CheY2 have 90% and 89% identity to CheY1 and CheY2 of *A. fabrum*, respectively [9,45]. In the experiment to explore the mechanism of function differentiation of two CheY paralogs, we find that the function of CheY2 is significantly improved by the substitution of some divergent key residues on the motor-binding surface for the corresponding residues of its paralogous CheY1, suggesting that protein function may be improved by the substitution of the divergent key residues in the functional domain for the corresponding residues of its paralogs.

The bacterial flagellar motor is ultrasensitive to the free CheY-P in the cytoplasm [49,50,51]. Without question, any slight change of the binding affinity of CheY for FliM will significantly affect the function of CheY in controlling flagellar motor. Therefore, structure resolution of *A. fabrum* CheY variants with enhanced function will be helpful for us to understand the mechanism of CheY to bind and regulate the flagellar motor. In addition, it was reported that the chemotactic regulator affected the surface attachment, biofilm formation, and pathogenicity of *A. fabrum* [52,53]. Consequently, two CheY2 variants that significantly enhance *A. fabrum* chemotaxis are worth further exploring to determine their effects on the pathogenicity of *A. fabrum*.

## 5. Conclusions

In conclusion, two CheY paralogs play significantly different roles in regulating the chemotactic response and swimming behavior of *A.*
*fabrum*. CheY2 is the main chemotactic response regulator of *A.*
*fabrum*. The mechanism resulting in the functional difference of two *A. fabrum* CheY paralogs is different from any previously reported mechanism and the genuine function of CheY1 is worth further investigation. The functional differentiation of two *A. fabrum* CheY paralogs mainly arises from the divergence of the key residues on the motor-binding surface and thus these key residues play a key role in distinguishing and improving their functions. The functional improvement of CheY2 by the substitution of some key residues on the motor-binding surface demonstrates that the reciprocal substitution of the divergent key residues in the functional subdomain between paralogous proteins may provide a new approach for rational protein engineering.

## Figures and Tables

**Figure 1 microorganisms-09-01134-f001:**
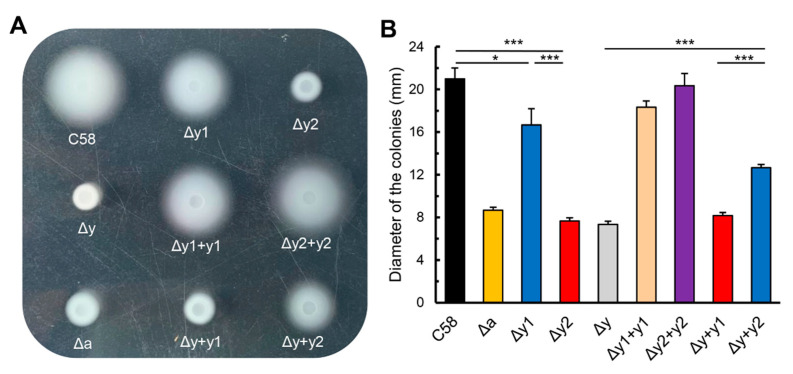
Chemotactic migration of *cheY*-deletion mutants and complemented strains of *Agrobacterium fabrum* in the swim plate. (**A**) Typical colonies of different *A. fabrum* strains after growing for 40 h on the swim agar plate. (**B**) Colony diameter of different *A. fabrum* strains after growing for 40 h on the swim agar plate. The data are the means of three independent replicates with standard deviations. The bars connected by “*” and “***” mark mean that they are different in a statistically significant manner at *p* < 0.05 and 0.001, respectively via the unpaired Student’s *t* test. C58, *A. fabrum* wildtype C58 strain; ∆a, *cheA*-deletion mutant; ∆y1, *cheY1*-deletion mutant; ∆y2, *cheY2*-deletion mutant; ∆y, *cheY1* and *cheY2* double-deletion mutant; ∆y1 + y1, ∆y1 mutant complemented with *cheY1* gene; ∆y2 + y2, ∆y2 mutant complemented with *cheY2* gene; ∆y + y1, ∆y mutant complemented with *cheY1* gene; ∆y + y2, ∆y mutant complemented with *cheY2* gene.

**Figure 2 microorganisms-09-01134-f002:**
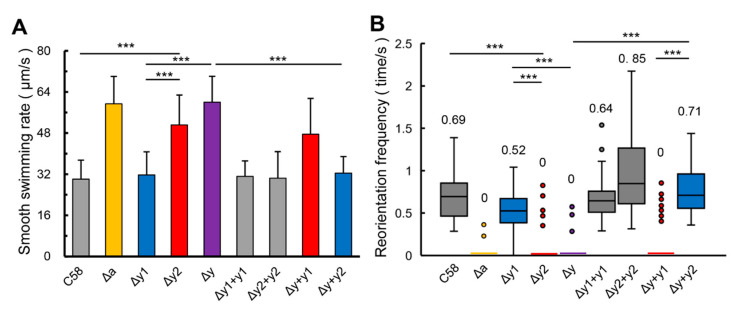
Effects of two CheYs on the free-swimming behavior of *A. fabrum* cells. The cells of different *A. fabrum* strains, which can express eGFP as the reporter protein for tracking their swimming tracks, were harvested at middle-logarithmic phase, washed, and resuspended in buffer at room temperature. Fifty cells of each strain were recorded by digital camera to analyze their swimming behaviors. (**A**) Average swimming speed (μm/s) of various *A. fabrum* strains. The bars connected by “***” mark means that are different in a statistically significant manner at *p* < 0.001 via the unpaired Student’s *t* test. (**B**) Tukey box plot of reorientation frequency (s^−1^) of various *A. fabrum* strains. Box displays the 25–75th percentile of the observations. The bars indicate the maximum and minimum value. The horizontal line within the box indicates the median, the value of which is also displayed above the respective box. The line in the horizontal axis represents that most of the observations are zero and dots represent outliers. The boxes connected by the “***” mark mean that they are different in a statistically significant manner at *p* < 0.0001 via the Wilcoxon–Mann–Whitney test. The strain names represented by the labels in the horizontal axis are the same as in Figure 1.

**Figure 3 microorganisms-09-01134-f003:**
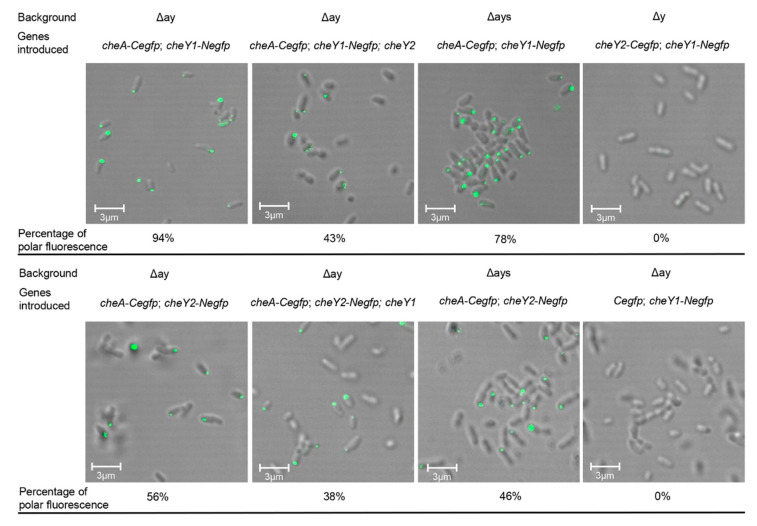
Polar co-localization of CheA with CheY1 or CheY2. eGFP was split into two non-fluorescent parts. The C-terminal part (C*egfp*) of eGFP was fused into the C-terminus of CheA (or CheY2) and the N-terminal part (N*egfp*) of eGFP was fused to the C-terminus of CheY1 (or CheY2). The introduced genes were equivalently co-expressed in the indicated *A. fabrum* strains. Interaction between two fusion proteins results in the reconstitution of two split-eGFP parts to emit fluorescence. Cells expressing the indicated fusion genes were observed by using confocal laser-scanning microscopy. The percentage represents the fraction of the cells with polar fluorescence. The background represents the host strains. The introduced genes indicate the genes complemented to the host strain by a plasmid. Δay: *cheA*, *cheY1* and *cheY2* triple-deletion mutant; Δays: *cheA*, *cheS*, *cheY1* and *cheY2* quadra-deletion mutant; Δy: *cheY1* and *cheY2* double-deletion mutant.

**Figure 4 microorganisms-09-01134-f004:**
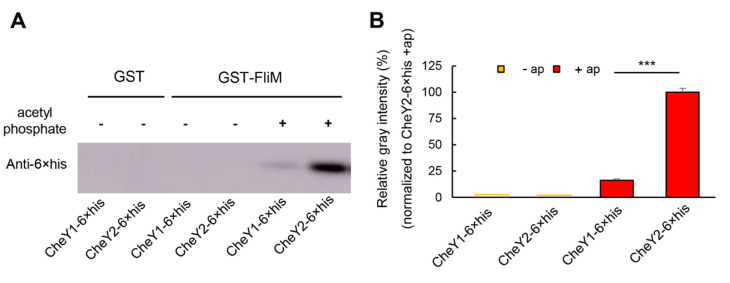
In vitro analysis of the interactions of two CheY proteins with FliM. Protein–protein interaction was tested by GST pull-down assay. The GST-FliM fusion protein was used to pull-down CheY1-6×his or CheY2-6×his fusion proteins. GST was used as the control. The pulled-down proteins were analyzed by Western blot using the anti-6×his tag antibody. (**A**) Western blot test of CheY1-6×his and CheY2-6×his pulled-down by GST or GST-FliM in the absence (−) or presence (+) of acetyl phosphate. (**B**) Relative gray intensities of the Western blot bands of CheY1-6×his and CheY2-6×his pulled-down by GST-FliM in the absence (−ap) or presence (+ap) of acetyl phosphate. The line in (**B**) represents the value close to zero. Data are the means of three biological replicates with the standard deviations. The bars connected by “***” mark mean that they are different in a statistically significant manner at *p* < 0.001 via unpaired Student’s *t* test.

**Figure 5 microorganisms-09-01134-f005:**
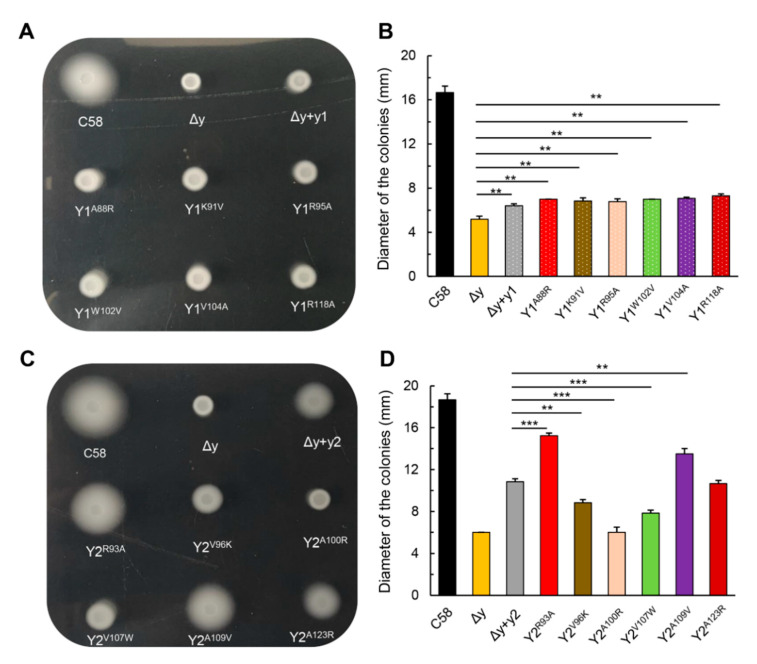
Effects of the single residue substitution of the key residues in the motor binding surface of CheY on the function of CheY in recovering the chemotactic migration of CheY-deficient mutant in the swim plate. Plasmids coding various CheY variants (or native CheYs) were respectively introduced into the *cheY1*-*cheY2* double-deletion mutant (Δy) to complement the function of CheY. Equal amounts of the cells from these *A. fabrum* strains were inoculated on the swim agar plate and grown for 40 h at 28 °C. (**A**,**C**): Typical colonies of these tested *A. fabrum* strains. (**B**,**D**): The diameters of these colonies were measured to assess the activities of these CheY variants in restoring the chemotactic response of *A. fabrum* to nutrients. Data show the means of three biological replicates with the standard deviations. The bars connected by “**” and “***” mark mean that they are different in a statistically significant manner at *p* < 0.01, and 0.001, respectively, via the unpaired Student’s *t* test. C58, *A. fabrum* wildtype C58 strain; ∆y, *cheY1* and *cheY2* double-deletion mutant; ∆y + y1, ∆y mutant complemented with *cheY1* gene; ∆y + y2, ∆y mutant complemented with *cheY2* gene; Y1^A88R^, Y1^K91V^, Y1^R95A^, Y1^W102V^, Y1^V104A^, and Y1^R118A^ represent ∆y mutant expressing the corresponding single residue-substituted CheY1 variants respectively; Y2^R93A^, Y2^V96K^, Y2^A100R^, Y2^V107W^, Y2^A109V^, and Y2^A123R^ represent ∆y mutant expressing the corresponding single residue-substituted CheY2 variants respectively.

**Figure 6 microorganisms-09-01134-f006:**
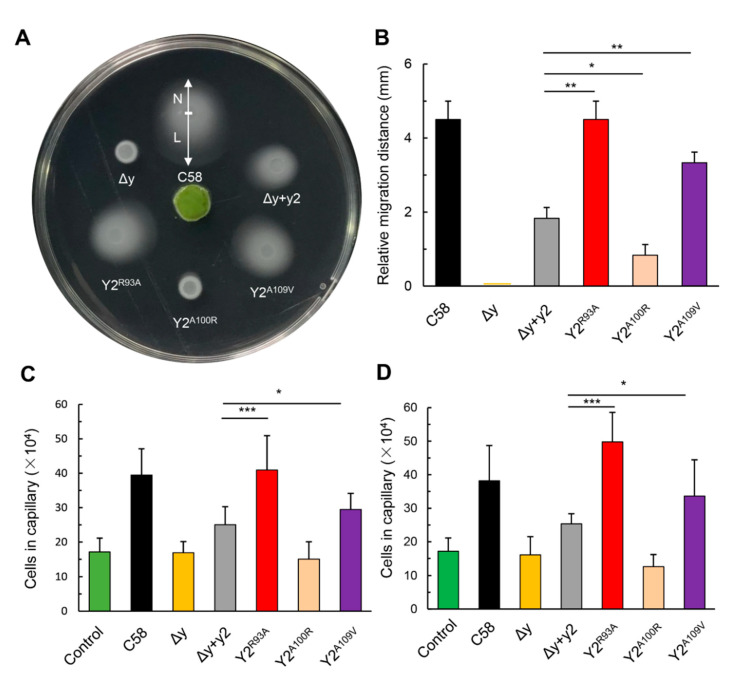
Effects of CheY variants on the chemotactic response of *A. fabrum* to host plant attractants and a single attractant. A leaf disc of kalanchoe was placed on the center of the plate to generate an attractant concentration gradient. Equal amounts of various *A. fabrum* cells were inoculated around the leaf disc with equal distance from the leaf disc and incubated for 40 h at 28 °C. The distance from the inoculating spot to the edge toward the leaf disc (L) minus the distance from the inoculating spot to the edge away from the leaf disc (N) was defined as the relative migration distance, which was used to assess the chemotactic response of *A. fabrum* to host attractants. (**A**) Typical colonies of the tested *A. fabrum* strains. (**B**) Relative migration distance of these strains. (**C**) Cells migrating into the capillary containing sucrose. (**D**) Cells migrating into the capillary containing AS. Data show the means of three biological replicates with the standard deviations. The bars connected by “*”, “**” and “***” mark mean that they are different in a statistically significant manner at *p* < 0.05, 0.01 and 0.001, respectively via the unpaired Student’s *t* test. The strains represented by the labels are the same as in Figure 5. Control indicates C58 cells migrating into the capillary containing buffer only.

## Data Availability

The data that support the findings of this study are available from the corresponding author upon reasonable request.

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
