# Peer review of "The Divergent Key Residues of Two *Agrobacterium fabrum* (*tumefaciens*) CheY Paralogs Play a Key Role in Distinguishing Their Functions"

_microorganisms, 2021, doi:10.3390/microorganisms9061134_

Round 1

Reviewer 1 Report

The manuscript is interesting, but  in this form is too hard to read. That because:

1) The methodologies descriptions from Materials and Methods is too long; 

2) At the major figures names, the authors give too much text. This must be put in manuscript body, not here. Statistically signification must be write once, at subchapter 2.10,  entitled  Statistical analysis; 

3) Between the names of the figures and the manuscript body must to exist a space;

4) The paper do not have a Conclusions. This can be done easily, putting all the the text from the end of the row 508 ( text  which begin with: ''Combining these results, we can conclude that the mechanism....) at the end chapter  entitled ''Conclusions'''

Reviewer 2 Report

Gao and collaborators in the manuscript entitled ,,Two Agrobacterium tumefaciens CheYs have different effects on chemotaxis and the divergent key residues play a key role in distinguishing their functions " submitted to Microorganisms (ISSN 2076-2607) undertook investigation of differences in the functional roles of two paralogous CheYs in an important plant pathogen Agrobacterium tumefaciens. The topic is worth scientific effort and the authors utilized complex molecular biology tools to answer their research hypotheses. I have some minor comments to this work:

- The title and abstract are too lengthy. They should be more concise. For instance: in the abstract there is an extremely long sentence: ,,The single respective replacements of key residues R93 and A109 on the motor-binding surface of CheY2 by alanine (A) and valine (V), the residues of the corresponding sites in CheY1, significantly enhanced the function of CheY2 in regulating the chemotactic response of A. tumefaciens CheY-deficient mutant Δy to nutrient substances and host attractants, concluding that the divergence of the key residues in the functional subdomain is the decisive factor of functional divergence of these two CheY homologs and protein function may be improved by the substitution of the divergent key residues in the functional domain for the corresponding residues of its paralogs.” without any commas or full-stops. Besides, the abstract should clearly state the aims of the study, the utilized methodology, collected outcomes and the conclusions.

  • I suggest to improve the language quality of the manuscript. The prepositions or plurals are missing, there are not enough commas, some sentences are definitely too long or clumsy. Unprofessional words appear (better, line 149) E.g. ,,The single respective replacements of key residues R93 and A109 on the motor-binding surface of CheY2 by alanine (A) and valine (V), the residues of the corresponding sites in CheY1, significantly enhanced the function of CheY2 in regulating the chemotactic response of A. tumefaciens CheY-deficient mutant Δy to nutrient substances and host attractants, concluding that the divergence of the key residues in the functional subdomain is the decisive factor of functional divergence of these two CheY homologs and protein function may be improved by the substitution of the divergent key residues in the functional domain for the corresponding residues of its paralogs.”; ,, Unlike E. coli, which has only one set of chemosensory system and encodes only one cheY for chemotaxis, many bacteria usually have more than one chemosensory system and multiple response regulator CheYs [9], but not all CheY paralogs participate the chemotactic signal transduction.”, ,, For example, Borrelia burgdorferi possesses two complete sets of core chemotaxis proteins and three CheYs, but only CheY3 is essential for motility and chemotaxis [10] and CheY2 is suggested to regulate a virulence determinant that is required for the infectious life cycle of Borrelia burgdorferi [11].”

  • The introduction section should be improved. There are some sentences that are trivial or lacking enough details for a scientific publication. E.g. ,,Chemotaxis is an environment-adaptive behavior of motile microbes, which allows the motile microbes to…”; ,,Some CheYs may be involved in other specific regulatory function.”. Broader review of the literature is needed. The functions of CheZ and CheS have not been discussed in the first section of the introduction part.

  • In general the materials and methods section should be enriched with more details, especially regarding the molecular biology methods. Some parts are not detailed enough. Just few examples:,, The solid medium is the same as the liquid, except for the extra agar.” (how much agar? Cannot be ,,the same”), ,,The desired A. tumefaciens mutants were screened by PCR and verified by DNA sequencing.”(not enough details); line 134 – how it was ,,normalized”?; ,,equal cells“(line 134) – you mean equal volume of cell suspension?; line 151 – what do you mean by ,,appropriate concentration”?; line 227 - ,,equal volumes of various samples”; line 230 – details on the device for chemiluminescence analysis are needed; line 235 – provide details on the DNAMAN software; line 239 – ,,traditional PCR process” not enough details; line 241 – please provide details on the plasmid introduction method;

  • Statistical analysis section needs improvement. Which statistical tests were conducted to decide whether to use Student t-test or the alternative non-parametric approach?
  • Results: In my opinion repetition of the methodology is unnecessary in the results section and underneath the figures. It would be better to improve descriptions in the materials and methods section. The collected results should be discussed in more detail. Here, they are a bit lost in the results section, as the methodology and introduction parts diffused there.

  • Results: Please comment why just some bars are connected by lines with asterisks of the statistical significance? Just these have been compared? The differences between the others are not statistically significant? This should be obvious for the reader. For me the statement ,, The asterisks of “**” and “***” represent P values of two compared strains < 0.01, and 0.001 via unpaired Student’s t test, respectively.” is not correct – it should be rather changed to ,,The bars connected by “**” and “***” mark means that are different in a statistically significant manner at P < 0.01, and 0.001, respectively via unpaired Student’s t test.”
  • Results: please avoid unnecessary repetitions of information that have already been provided in the introduction section e.g. ,, Motility is an indispensable prerequisite for bacterial chemotaxis. Motile bacteria exhibit various motility patterns [42]. Chemotactic bacteria accomplish chemotaxis by regulating their swimming pattern to move toward favorable environment. The chemotactic response regulator CheY, after phosphorylated, can modulate flagellar motor activity, changing the swimming pattern [43].”
  • For better readability please provide intervals after figure legends
  • Results: Please be more specific with some statements e.g. ,, Results of the CheY-complemented strains further verify that…”
  • Could the authors comment why of 8 key resides 6 have been selected for further study?
  • Discussion: In my opinion it should be broader, with more literature data. There is too much focus on repetition of the results.
  • Some editorial work is needed: omitted italics (e.g.lines 396, 397), commas etc
  • Why the supplement has been uploaded twice?
